# The Rice Alpha-Amylase, Conserved Regulator of Seed Maturation and Germination

**DOI:** 10.3390/ijms20020450

**Published:** 2019-01-21

**Authors:** Rebecca Njeri Damaris, Zhongyuan Lin, Pingfang Yang, Dongli He

**Affiliations:** 1Key Laboratory of Plant Germplasm Enhancement and Specialty Agriculture, Wuhan Botanical Garden, Chinese Academy of Sciences, Wuhan 430074, China; njerirebecca09@gmail.com (R.N.D.); lzygry@sina.com (Z.L.); 2University of Chinese Academy of Sciences, Beijing 100049, China; 3School of Life Sciences, Hubei University, Wuhan 430070, China

**Keywords:** alpha-amylase, rice, seed germination, classification, expression regulation

## Abstract

Alpha-amylase, the major form of amylase with secondary carbohydrate binding sites, is a crucial enzyme throughout the growth period and life cycle of angiosperm. In rice, alpha-amylase isozymes are critical for the formation of the storage starch granule during seed maturation and motivate the stored starch to nourish the developing seedling during seed germination which will directly affect the plant growth and field yield. Alpha-amylase has not yet been studied intensely to understand its classification, structure, expression trait, and expression regulation in rice and other crops. Among the 10-rice alpha-amylases, most were exclusively expressed in the developing seed embryo and induced in the seed germination process. During rice seed germination, the expression of alpha-amylase genes is known to be regulated negatively by sugar in embryos, however positively by gibberellin (GA) in endosperm through competitively binding to the specific promoter domain; besides, it is also controlled by a series of other abiotic or biotic factors, such as salinity. In this review, we overviewed the research progress of alpha-amylase with focus on seed germination and reflected on how in-depth work might elucidate its regulation and facilitate crop breeding as an efficient biomarker.

## 1. Rice Seed Maturation and Germination

### 1.1. Rice Seed Maturation and Starch Storage

Rice is a staple food for more than half of the world’s population (www.fao.org). Rice seeds are enriched with starch, proteins, and lipids. During rice seed maturation, double fertilization results in a diploid embryo and a starchy triploid endosperm, with the embryo consisting of an embryonic axis surrounded by a single cotyledon called scutella. Embryonic shoot apical meristem differentiates three days after fertilization (DAF) and leaf primordium emerges at 5–8 DAF; meanwhile, the cell wall of endosperm forms at 3–4 DAF, endosperm cellularization and endoreduplication complete at approximately 6–10 DAF; the seed organs enlarge and mature at 11–20 DAF and the endosperm accumulates the storage reserves and then the seed triggers desiccation at about 30 DAF (Figure 1A). As a result, only the embryo and aleurone are alive in the matured seed [1]. 

The rice grain assimilates approximately 30% of its carbohydrate from non-structural carbohydrates in leaf sheaths and culms before heading which is replaced by the photo-assimilated carbon after heading [2,3]. A clear relationship was found between the enzyme activity of alpha-amylase (EC 3.2.1.1) and starch content in the leaf sheath after heading with the activity being consistent with the degree of starch degradation [2]. High alpha-amylase expression and activity was detected in high-temperature-triggered grain chalkiness [4]. 

Starch accumulates in the endosperm as two d-glucose homopolymers including amylose with only linear α-1,4-glucan and amylopectin with α-1,6 glucan linked branches [5,6]. Several enzymes are involved in rice seed starch synthesis, including sucrose ADP-glucose pyrophosphorylase, phosphoglumutase, starch synthase, starch branching enzyme, and starch debranching enzyme. Starch occurs either as soluble starch, sucrose or sucrose derivatives such as fructosyl-oligosaccharides (fructans), or insoluble starch. During the early phase of seed development, water-soluble carbohydrates starch can be stored [7]. While in a tropical cereal species that lacks the fructan accumulation enzyme [8], accumulation of starch occurs in the form of insoluble starch grains. Starch grains are transparent in nature and can be classified as simple or compound starch grains. Starch grain is formed in amyloplast, a double-membrane bound organelle that is involved in starch synthesis and storage [9]. Compound starch granules are assembled by smaller starch granules into 10–20 µm particles [10,11]. The accumulated starch in maturing seeds equips them with the sufficient substrates for seed germination.

### 1.2. Seed Germination

Seed germination is a critical process ensuring the continuity of life in plants that depend on it as the exclusive mode of propagation [12]. Effective seed germination would result in robust and healthy seedling development, with seedlings that have the capacity to withstand biotic or abiotic stresses and develop into high yielding crops [13]. Seed germination (Figure 1B) requires conducive environmental conditions, including water, oxygen, and temperature [14], and a combination of numerous cellular processes, such as synthesis, transport, signaling, sensing, and hydrolysis. Germination begins with the imbibition of water and ends with the protrusion of the coleoptile and radicle [12]. Rice seed imbibition is a tri-phasic process with phase (I) characterized by rapid water uptake and DNA repair (about 0–24 h after imbibition, HAI) [15,16], followed by a plateaued phase (II) that was characterized by mitochondrion synthesis [17] and the translation of stored mRNA (24–48 HAI) [18], and a final increase in water uptake, which occurs after the embryonic axes have exited the structures surrounding it (48–72 HAI, phase III) [12,19] (Figure 1 A). The quiescent dry seed rapidly resumes metabolic activity upon imbibition, resulting in the degradation of macromolecules stored in the endosperm which, in turn, supply the energy and nutrients that are needed by the embryo for seedling establishment. 

During rice seed germination, sucrose is rapidly consumed by the growing embryonic axis. The dry embryo stores carbohydrates in the form of sucrose and the shoot tissues consume the largest proportion of the sucrose available in the embryo [22]. The dry embryo contains few starch granules and the sugars assimilated by the embryo are transiently converted to starch. About 24–36 h after imbibition, numerous larger granules accumulate around the vascular tissues, scutella, and epithelium [22], making phase II perhaps the most critical stage of rice seed germination. The mechanism by which the soluble sugars are transported to the embryo from the degraded reserves in the endosperm remains unknown, however there must be frequent communication, transport, and signaling between the embryo and the endosperm, all aimed at establishing a successful germination. In barley, peptide transporters are suggested to attribute to these bi-directional interactions [23,24]. 

The major source of energy for seed germination and seedling establishment is the degradation of stored starch in the endosperm. Complete starch hydrolysis is achieved by the combined action of α-amylase, β-amylase, debranching enzyme, and α-glucosidase. Alpha-amylase (1,4-α-d-glucan maltohydrolase) is a calcium metalloenzyme and it acts at random sites along the starch chain giving rise to α-maltose and α-glucose [25], while β-amylase (1,4-α-d-glucan maltohydrolase) works from the non-reducing end of starch hydrolyzing the second α-1,4 glycosidic bond and cleaving off one maltose unit at a time [26]. Alpha-amylase is synthesized de novo during seed germination in the presence of endogenous GA (gibberellic acid) from the embryo [27]. Conversely, β-amylase is present prior to germination in an inactive form without control of GA [26], which is activated by protease activity that cleaves its carboxyl terminus [6,27,28]. Synthesis of β-amylase occurs only in the aleurone layer, while that of alpha-amylase occurs both in the aleurone layer and the epithelial layer. In our previous study, we detected one of the alpha-amylase (*amylase1A*) in the rice seed embryo and demonstrated that the levels of its protein and mRNA both significantly increase and peak at 48 HAI during germination [29], which in addition to its responsiveness to phytohormones, may determine the starch granule transformation in the germinating rice seed.

## 2. Functions of Different Rice Seed Organs in Germination

### 2.1. The Embryo

For the endosperm-dominant rice seeds, the embryo makes up roughly 2% of the total dry weight of the seed while the endosperm accounts for 98% of the total dry weight of which 87.8% is the starchy endosperm [30]. Despite its relatively low proportion, the embryo plays a principal role in the plant life cycle since it contains all of the genetic information that is required for the initiation of seed germination. A dry rice seed contains more than 17,000 mRNAs [31] and long-lived storage mRNA analyses show that a 2.4-fold preferential accumulation of genes is involved in biological roles in the embryo compared to those in the endosperm [30]. The over-represented embryonic-favored genes are related to “ribosome biosynthesis”, “translation”, “rRNA binding”, “ribosomal large and small sub units”, and “structural constituents of ribosome” [30], which were considered together to reveal the readiness of the embryo to initiate translation. Even though newly synthesized mRNA facilitates rapid and even germination, mRNA stored in a matured embryo provides the requisite proteins for the initiation of germination. This was demonstrated by blocking transcription by α-amanitin or actinomycin D, which did not affect germination, while blocking translation with cycloheximide halted germination [29,32,33] which confirmed that translation is a more critical and conserved process than transcript for seed germination [34].

### 2.2. The Endosperm

The endosperm is the starch-rich component (about 70%) of the cereal seed that dies during seed maturation and desiccation and is surrounded by the living aleurone layer. Although considered dead, the endosperm still participates in various metabolic activities such as redox activities that are crucial during germination [35]. The endosperm also stores proteins (about 10%), oils, and amino acids. The proteins are reduced during germination by the action of proteases, providing the nitrogen that is required by a germinating seed. In addition, the endosperm stores the inactive form of β-amylase which is activated by the secreted proteases [26].

### 2.3. The Scutellum and Aleurone Layer

The scutellum, a single layer of epithelial cells (cotyledon of monocots), separates the embryo from the endosperm. Its thin layer with a high surface area facilitates the absorption of nutrients from decomposing endosperms to embryo during germination and some unidentified protein transporters in it will facilitate this bidirectional interaction. The aleurone layer (peripheral endosperm), which is the part of the endosperm that remains active during seed maturity, is densely packed with cuboid cell structure. However, at the last phase of seed germination, the aleurone layer undergoes programed cell death. It contains aleurone proteins and oil bodies. It is known that the GA that is synthesized in the germinating embryo is released to the aleurone layer via the scutellum where it triggers the release of hydrolytic enzymes, including alpha-amylase, by utilizing its stored reserves that are eventually released into the endosperm [6,36].

## 3. Classification of Alpha Amylase

Alpha-amylase has been placed in family 13 of the glycosyl hydrolases and is found in archaea, bacteria, plants, and animals [37,38,39]. Cereal alpha-amylase genes contain small localized insertions, termed “alpha-amylase signatures”, since they specify the gene subfamily to which a particular alpha-amylase belongs. There are two signature regions that have been identified in the cereal alpha-amylase, namely, signature I and signature II; the signature II domain sequence is highly conserved. Huang et al. [40] further classified alpha-amylase into two large groups accordingly, AmyA and AmyB, with the former having 6-or 9-bp insertions (Amy1 (TCC GGG), Amy2 (TCC GGC CAT), and Amy3 (CAA/G GCN)) in their sequence. Based on the polymorphic sequence of the signature I region, the *AmyA* is divided into *Amy1* and *Amy2* [40]. Consequently, alpha-amylase genes in cereals are classified into three subfamilies, including *Amy1*, *Amy2*, and *Amy3(AmyB)* [41].

Isoelectric focusing analyses on germinating rice seed extracts have identified at least three alpha-amylase isozymes [42,43]. Four additional immunological cross-reactive products were produced by in-vitro translation of poly(A)^+^ mRNA from isolates of germinating rice seeds [42]. Therefore, there are approximately eight functional alpha-amylases in the rice genome [44,45], that is *Amy1A, Amy1C*, *Amy2A*, *Amy3A*, *Amy3B*, *Amy3C*, *Amy3D,* and *Amy3E.* Alpha-amylase isozymes in barley are classified into type A (low pI) and type B (high pI), with at least two isozymes in each type [46]. The alpha-amylase genes in wheat consist of three subfamilies [47] and analyses of alpha-amylase in germinating wheat seeds revealed 27 isozymes [48].

Analyses of the genomic clones of rice alpha-amylase have revealed 10 distinct genes that are classified into five hybridization groups and three subfamilies, namely subfamily RAmyl (genes *RAmylA-B*), Group 1; subfamily RAmy2 (gene *RAmy2A*), Group 4; subfamily RAmy3 (genes *RAmy3A-C*), Group 3; subfamily RAmy3 (gene *RAmy3D*), Group 2; and subfamily RAmy3 (gene *RAmy3E)*, Group 5 and *RAmy3F* was placed in group 6 because it was not reported to be among any group in the literature that was read [44,49] (Table 1).

## 4. Structure of the Rice Alpha-Amylase Gene

The gene sequence of the 10 rice alpha-amylases listed in Table 1 were downloaded from the rice annotation project (http://rice.plantbiology.msu.edu/) and the gene structure was drawn using the Gene Structure Display Severer 2 [55]. Their exon numbers range from 3 to 11 with most containing 3–4 exons, while *RAmy3E* contained the most. *RAmy3B* has a notably long intron of 8 kb. All *RAmy1*(A/B and C) gene structures exhibited the greatest similarity. *RAmy1B* had the same size of exon and introns as *RAmy1A* and only the *RAmy1C* structure exhibited a small additional exon. Notably, *RAmy3E* lacks the 3′ downstream untranslated region, while *RAmy3F* has the longest 5′ upstream untranslated region (Figure 2A).

We further analyzed the evolution of the alpha-amylase in the plant. The alpha-amylase protein sequence of rice, and two other model plant barley (6 proteins), and *Arabidopsis* (3 proteins) were downloaded and aligned by MEGA version X [56] (Appendix A). The evolution tree was inferred using the Neighbor-Joining method. All of the 19 alpha-amylase proteins were classified into four major clades, with *RAmy3A, B, C,* and *D* being clustered remotely to the others. *RAmy1C*, 3E, and 3F share the most recent ancestor with Arabidopsis. This is consistent to the conclusion by Huang et al. [40] that both monocots’ and dicots’ alpha-amylases share a common ancestor (Figure 2B).

Ochiai et al. [57] described a detailed crystal structure of *RAmy1A* (Figure 2C), showing that it contains 412 amino acid residuals with three domains including an N-terminal (β/α)_8_-barrel domain (domain A, amino acid residues 1–124 and 171–375), a long-looped domain (domain B, amino acid residues 125–170) inserted into domain A, and a C-terminal β-sheet domain (domain C, amino acid residues 376–428). The structure consists of 11 α-helices and 15 β-strands. Domain A contains three invariant catalytic residues (Asp-Glu-Asp) and calcium ions that contribute to protein fold stabilization. The alpha-amylase family have a secondary carbohydrate binding site termed the SBS1 in the N-terminal and SBS2 in the carboxyl-terminal that is situated on the surface that is far from the enzymes’ active site. SBS1 and SBS2 act synergistically in degradation of starch granules, a conclusion that was derived from the observation that site-directed mutagenesis of amino acid (Trp^278^ and Trp^279^) stacked onto adjacent ligand glucosyl residues at SBS1 and that those (Tyr^380^ and His^395^) that make numerous ligand contact at SBS2, inhibited the binding of the starch granule to the enzyme. Together, SBS1 and SBS2 direct the enzyme to the location on the starch granule with free alpha-glucan chain for catalytic action, leading to an efficient degradation of starch [58].

Sequence analysis indicated that the promoters of most alpha-amylases contain the conserved *cis*-factors such as gibberellin-response complexes (GARC) that were identified via functional analyses of the promoter region of barley low and high pI alpha-amylase [59,60]. GARC contains three conserved sequences that are found in all GA-responsive cereal alpha-amylase genes including W box (TTGACC/T) pyrimidine box (C/TCTTTT), GARE box (C/TAACC/GG/AA/CC/A), and TA box (TATCCA) [61]. *Cis*-acting elements responding to GA in *RAmy1A* promoter were searched and within the first 500 bp upstream region, four key elements of GARC were found (Figure 2D). Mutation in GARE results in the loss of GA-responsiveness suggesting that GARE plays a pivotal role in GA signaling [59]. The low pI alpha-amylase promoter in barley and wheat contains a TAACAGA, a TAACAAA-like box of GARE, that functions like GARE, and mutation in the sequence renders the promoter insensitive to GA [62]. Apart from the GARC, many alpha-amylases promoters can respond to sugar with the sugar response complex (SRC), which includes the GC box (cTACGTGGCca), the G box (CTACGTGG), and the TA box [52,63]. The shared TA box between GARC and SRC indicated that the GA and sugar may competitively regulate the alpha amylase expression.

## 5. Expression Traits of Alpha-Amylase 

Using the RiceXPro database version 3.0 (http://ricexpro.dna.affrc.go.jp/), the expression of different alpha-amylase isozymes was checked in the entire life cycle of rice. Apart from *RAmy3B*, of which its expression data was not available, information on the rest of the genes was obtained. The expression graphs were placed into three groups depending on expression sites. They included those that were expressed highly only in developing seeds such as *RAmy1A, 1B, 3A*, and *3C,* however not in other tissues or other stages of growth. *RAmy1A* is highly expressed in the ovary and embryo exclusively in the first 48 DAF with the highest expression at 10 DAF in the embryo. *RAmy3A* is only expressed in the endosperms of maturing rice seeds at 42 DAF. High expression levels of the alpha-amylase genes in immature ripening seeds could help transform the transiently stored starch to sugar and rapidly satisfy the requirement of energy and substrates (Appendix A), especially at the early stage of embryo maturation. In wheat, some *Amy* genes are exclusively expressed in immature grains too, including wheat *α-Amy3* [65]. The second and third categories consist of genes that are expressed in all of the tissues during all of the stages of development, which only vary in their degree of expression. They include *RAmy1C, 2A, 3D, 3E*, and *3F* with *RAmy3E* and *RAmy3F* exhibiting the highest expression in all of the tissues and stages of rice growth (Appendix A). 

During rice seed maturation, starch accumulated in the leaf sheaths through photosynthesis is transported to the sink constantly after degradation either by the hydrolyzing enzymes or by phosphorolytic action. As rice head, *RAmy2A* increases in the leaf sheath [66] (Appendix A, Table 1), which will directly affect the yield. Compared to a normal yielding variety, a higher yielding cultivar accumulated higher *RAmy2A* levels [2]. While in wheat, *α-Amy1* gene is responsible for the late maturity alpha-amylase (LMA), with GA playing a crucial role in its expression [67], which will result in the low falling number. 

In the germinating rice seed, different subfamilies show preferential expression either in the embryo or in the aleurone layer. *RAmy3B-C* and *RAmy3E* are preferentially expressed in the aleurone layer with *RAmy3C* abundance being three times higher than that of *RAmy3B* [49] (Table 1). Although *RAmy1A-B* and *RAmy3D* are detected in both the embryo and the aleurone layer, they exhibit varying accumulation rates with *RAmy1A-B* having the highest accumulation level during germination [49]. According to Karrel et al. [49], *RAmy1A* mRNA is the most dominant alpha-amylase transcript at 6 days post germination, while *RAmy3D* is transiently expressed in the early germination stage in the embryo, suggesting its involvement in the initial starch hydrolysis in the scutella region. Consequently, *RAmy3D* gene, which is the first alpha-amylase gene to be expressed during seed germination, is detectable 12 HAI, attaining its peak at 24 h, and its expression is turned off at 48 h, demonstrating its critical role in initiating seedling development. This could potentially explain why *RAmy3D* and *RAmy3E* are metabolite regulated genes exhibiting the highest levels of expression under sugar starvation conditions [68,69]. Hydrolyzing enzymes are also expressed in the endosperms of maturing rice seeds (Appendix A), with *RAmy1A* and *RAmy3C* being expressed in the internal (basal dorsal) sections of developing endosperms, while the sugar regulated *RAmy3D* is confined to the region near the ventral aleurone [51] (Table 1).

The major site for alpha-amylase expression has always been thought to be the aleurone layer in germinating rice seed [70]. However, it was demonstrated that alpha-amylase is initially expressed in the scutella’s epithelial tissue in starchy seeds and is later expressed in the aleurone layers at high levels [44], and both tissues can secrete alpha-amylase subsequently into the starchy endosperm. This further indicates the bidirectional interaction between the embryonic and endospermic tissues during seed germination and seedling establishment. The expression of alpha-amylase is triggered by the active GA at the transcription level through inducing positive trans activating factors for alpha-amylase [71]. Once in the scutella and aleurone layer, the GA initiates the synthesis of hydrolytic enzymes, including the alpha-amylase, which are secreted to the starchy endosperm. In the endosperm, alpha-amylase catalyzes the 1,4-α endo-glycolytic cleavage of the amylose and amylopectin, the key constituents of starch granules in plant cells [40], with the resulting carbohydrates, mainly sucrose, being shuttled back to the embryo to promote the mRNA translation and protein synthesis during seed germination and subsequent seedling establishment [23,30,72,73,74].

## 6. Regulation of *Alpha-Amylase* Expression

### 6.1. Hormonal Regulation of Alpha-Amylase

Regulation of seed germination by phytohormones has revealed two antagonistic hormones, ABA and GA, with GA promoting seed germination [60,74] and ABA inducing and maintaining seed dormancy [75] (Figure 3). Alpha-amylase gene expression is also regulated by these two phytohormones, where GA is a positive and ABA is a negative regulator. Direct crosstalk between GA and ABA has also been demonstrated, with ABA catabolic gene *ABA89OH-1* being positively regulated by GA in the aleurone layer, which potentially explains the low ABA levels in aleurone cells [67]. The protein OsPP2C51, a phosphate 2C clade A member and a major responsor in the ABA signaling cascade, positively regulates seed germination by enhancing alpha-amylase expression [76]. The resumption of respiration in the imbibed seed results in substrate utilization and substrate deficiency over time [77] which triggers the embryo to produce phytohormones, particularly GA. The expression of *OsGA3ox2*, the gene that is involved in the conversion of inactive GA to active GA, increases rapidly following imbibition, suggesting its essential role in de novo GA biosynthesis in the embryo, which in turn induces *RAmy1A* expression in the aleurone layer [60,78].

The treatment of the germinating embryo with GA biosynthesis inhibitor resulted in lower *RAmy1A* expression in the endosperm, indicating that de novo synthesis GA in the embryo is a prerequisite for the induction of *RAmy1A* in the aleurone layer of the endosperm [78]. Analysis of gene expression during the conversion of inactive GA to active GA, and using mutants defective in shoots and scutella formation, revealed that the scutella region of the embryo is essential for the induction of alpha-amylase [78]. Studies suggest that GA is perceived in the plasma membrane by gibberellin-binding protein (GBP) [79,80,81,82] and inside the cell by GID1 [83]. However, Kenji et al. [83] showed that GBP does not exist and that the GID1-DELLA system is solely responsible for GA reception in rice aleurone cells using a *gid1* mutant. The GA-GID1 complex binds to the repressor domain of DELLA (SLR1 and SLN1 homologues in rice and barley, respectively). This three-component complex, GA-GID1-SLR1/SLN1, in turn binds to an F-box protein, the SCF^GID2^ ubiquitin ligase, which targets SLR1/SLN1 for degradation by the 26S proteasome and the GA-inducible genes are expressed [14,84,85].

MYBs are a group of transcription factors with a conserved DNA binding domain that consist of three imperfect repeats made up of ~51 to 52 amino acids that are designated as R1, R2, and R3 [86]. GAMYB/MYBGA (R2R3 type) is a classic GA responsive alpha-amylase transcription factor of which its protein binds to GARC of alpha-amylase, positively regulating the expression of the gene in the aleurone layer [59,60]. GAMYB, together with the pyrimidine box-binding protein, OsDOF3, stimulates the promoter activity of RAmy1A [60] (Figure 3). The loss of function in a mutant of the gene, *gamyb,* resulted in the halted induction of alpha-amylase, indicating that MYBs are essential for the expression of alpha-amylase [87].

In the absence of GA, SLRI (DELLA protein) binds to GAMYB or other MYB TFs, resulting in the inhibition of alpha-amylase, while in the presence of GA, SLRI is degraded through GID2 and GAMYB or other GA responsive MYBs, which in turn facilitate alpha-amylase and other hydrolytic enzyme synthesis expression [83]. Itoh et al. [88] demonstrated that when cereal grains germinate, the aleurone expression of *RAmy1A* occurs on day two, increases on day five, and exhibits the highest expression on day six under GA control. Several transcription factors that bind to the TATCCA box of the alpha-amylase promoter sequence have been identified. They include *OsMYBS1*, *OsMYBS2*, and *OsMYBS3* [89].

### 6.2. Metabolite Regulation of Alpha-Amylase

Without GARE in the promoter, *Amy3D* and *Amy3E* genes exhibit limited responses or are insensitive to both GA and ABA [53,69]. The CGACG region of the Amy3D promoter and the G-box element identified by Hwang et al. [53] are involved in the metabolic regulation of the gene. During seed germination, the stored starch is hydrolyzed into simple sugars and is transported into the embryo to facilitate seedling establishment, which then repress the GA biosynthesis in the embryo [90,91]. Sugars and hormones tightly regulate seed germination so that sugars control GA biosynthesis in the embryo and alpha-amylase expression in the aleurone layer, further controlling the degradation of the starch stored in the endosperm [90]. Consequently, alpha-amylase can be induced by GA and sugar depletion while being repressed by ABA and sugar [90,91,92], a phenomenon that is also observed in cultured rice cells [92,93,94].

Sugar starvation can activate the alpha-amylase promoter through binding the TA box, a key element of the SRC [63]. The *α-Amy3* and *αAmy8* genes are highly induced under sucrose starvation in cultured rice suspension cells [95] and the mechanism of sugar repression involves their transcription rate and mRNA stability [94,95,96,97]. The promoter activity of *α-Amy8* is repressed by sucrose in the embryo, however it is activated by GA and is insensitive to sucrose repression in the endosperm, revealing a tissue-specific dominance that is regulated at transcription level [77]. The SRC of *α-Amy3, a* 105-bp, contains all the elements of SRC with two tandem repeats of the TA box and functions as the transcriptional enhancer of sugar depletion–induced promoter activity [52,63]. However, for the *α-Amy8* promoter, the 230-bp SRC/GARC contains a putative GC box, a GARE, and one TA box [77], functioning as a sucrose depletion and GA-induced promoter transcriptional enhancer [52]. The sugar starvation signaling pathway involves CIPK (CBL interacting protein kinase) proteins and act like calcium (Ca ^2+^) sensors that eventually trigger the accumulation of the global energy sensor factor, SNF1-relatedkinase 1A (SnRK1A) protein kinase, which in turn enhances the MYBS1-TA box interaction resulting in the activation of alpha-amylase transcription [54,98,99,100] (Figure 3).

In the aleurone cells, the MYBS1 is predominantly located in the nucleus under sugar starvation conditions. MYBS1 can bind to the SRC with GAMYB as its stable retention in the nucleus under sugar starved conditions [21]. However, under sugar availability conditions, the MYBS1 is transported out of the nucleus into the cytoplasm [21]. In addition, GAMYB and MYBS1 interact physically and synergistically to co-activate the αAmy8 GARC with the resulting bipartite protein (MYBS1-MYBGA)-DNA (GARE-TA box) complex being able to hold the MYBS1 in the nucleus and preventing it from being shuttered to the cytoplasm, eventually inducing the expression of alpha-amylase irrespective of sugar availability. Similarly, since GAMYB is also capable of nuclear export when all the nutrients are available, MYBS1 promotes its nuclear localization, implying that the bipartite MYB-DNA complex confines GAMYB in the nucleus [21]. This results in the sustained expression of hydrolases, particularly alpha-amylase [21], and in turn, to degrade the stored substrates for vigorous seed germination.

### 6.3. Other Factors Influencing the Expression of Alpha-Amylase

Abiotic factors such as high temperature and salinity reportedly affect the expression or regulation of alpha-amylase [51,101]. These may in turn halt seed germination, slow, or delay germination (Figure 3). The expression of alpha-amylase is down-regulated under copper stress conditions [102], while arsenic stress reduces alpha-amylase activity in wheat germinating seeds [103]. However, when wheat seeds germinate under anoxic conditions, the expression of *RAmy3D* is induced to provide the energy that is required for coleoptile elongation [104]. The expression of alpha-amylase is also inhibited by benzoxazinoids, a compound that is produced during the degradation of wheat residues. Therefore, benzoxazinoids could greatly affect rice germination in a previously wheat growing area [105]. Salinity has also been reported to inhibit seed germination via the reduction of alpha-amylase activity which occurs as a result of decreased bioactive GA content and of which its inhibition can be rescued by exogenous application of GA [101]. Chemicals, including maleic hydrazide, eugenol, and uniconazole, reduce the expression of alpha-amylase, especially *RAmy3B* and *RAmy3E*. It is an agronomic attribute that is employed by farmers to mitigate pre-harvest sprouting [50].

## 7. Challenges and Perspectives

Alpha-amylase is a crucial enzyme throughout the growth period and life cycle of rice and the expression of some alpha-amylase isozymes has been observed in all stages of development (Appendix A, http://ricexpro.dna.affrc.go.jp/). Alpha-amylase is critical during seed germination for the hydrolysis of stored reserves and provision of energy to the developing embryo. Overexpression of a transcription factor regulating the expression of alpha-amylase, OsMybcc-1, could result in seeds with enhanced germination compared to its mutant and wildtype (unpublished data). There is a considerable amount of literature on the measurement of amylase activity that is based on the crude extracts of the samples being analyzed (summarized in [106]) with the DNS method being the most common due to its simplicity. In the rice germinating seed, the amount of expressed alpha-amylase is high enough to be detected by these methods. However, these methods are usually tedious especially on buffer and reagent preparation. If these methods could be advanced and made into simple easily usable kits, such as the pH strips, they could facilitate the identification and detection of seeds with potential robust germination based on high alpha-amylase content, thus being utilized as biomarkers. This could be distributed to farmers and utilized for pre-sowing analysis and to choose seeds with potential high vigor.

However, higher accumulation of alpha-amylase during grain development, especially the filling stage, could result in rice grains with reduced quality, chalky, and with few stored starch grains [107]. Such grains would have low market value, negatively affecting farmers’ income and food availability. In wheat, the high accumulation of alpha-amylase during maturation coupled with high moisture availability leads to pre-harvest sprouting [108]. Dough from such seeds has a poor consistency and, in turn, baked products of low flavor and likability [109]. Therefore, it is a challenge for growing rice to regulate the appropriate amount of alpha-amylase at the right stage of development. For a researcher, therefore, the question is how can the effect of alpha-amylase during seed maturation be hindered considering that over expression of the alpha-amylase gene or its regulators enhances seed germination? How do we strike a balance between the appropriate levels at both seed maturation and germination? These questions could be resolved with time and following in-depth research. However, the environment presents other factors that influence gene expression, seed germination, and growth.

## Figures and Tables

**Figure 1 ijms-20-00450-f001:**
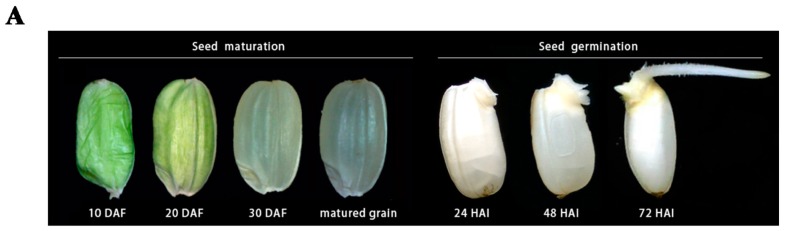
Rice Seed Maturation and Germination. (**A**) Changes in morphology during rice seed development [20] and germination (DAF (days after fertilization), HAI (hours after imbibition)). Rice seeds develop from the florets of the main stem which become the immature grains (the initial green color of the developing seed). The mature seed is tightly packed with starch granules (the translucent color) while imbibition rehydrates the starch granules, making them loose and thus appearing white as a result of light refection. (**B**) Events leading to rice seed germination showing the interaction and roles of different tissues during germination (Modified from Hong et al. [21]) (H_2_O (water), GA (gibberellic acid), TFs (Transcription factors)).

**Figure 2 ijms-20-00450-f002:**
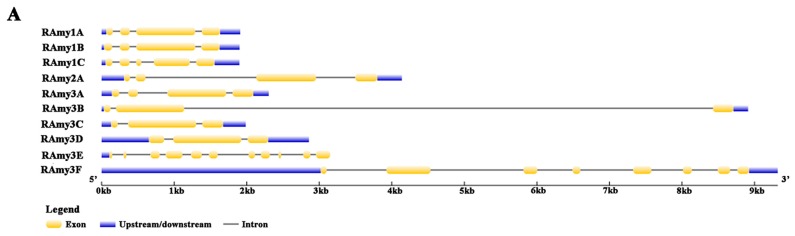
(**A**) Gene structure of the 10 rice alpha-amylase genes. (**B**) Phylogenetic relationship between alpha-amylase isozymes in rice, barley (AMY1.1-AMY1.6), and Arabidopsis (ATAMY(1,2 and 3)) by the Neighbor joining method drawn using MEGA version X [56]. The genomic and protein sequences were obtained from the Rice Genome Annotation Project using LOC_Os02g52700, LOC_Os02g52710, LOC_Os01g25510, LOC_Os06g49970, LOC_Os09g28400, LOC_Os09g28420, LOC_Os08g36900, LOC_Os08g36910, LOC_Os04g33040, and LOC_Os01g51754 [64] for *RAmy1A*, *RAmy1B*, *RAmy1C*, *RAmy2A*, *RAmy3A*, *RAmy3B*, *RAmy3C*, *RAmy3D*, *RAmy3E*, and *RAmy3F*, respectively. The barley gene ID that was used for downloading the sequences was obtained from [64] and the sequences were downloaded from NCBI. For Arabidopsis, the sequences were downloaded from TAIR. (**C**) The overall crystal structure of Amy1A. Calcium ions are represented as orange balls, secondary carbohydrate binding site (SBS1 indicated by the black arrow) comprising of oxygen atoms (red balls) and carbon atoms (deep blue shape), (cited from [57]) (**D**) Schemes of Constructs for the representative promoter of *Amy1A*.

**Figure 3 ijms-20-00450-f003:**
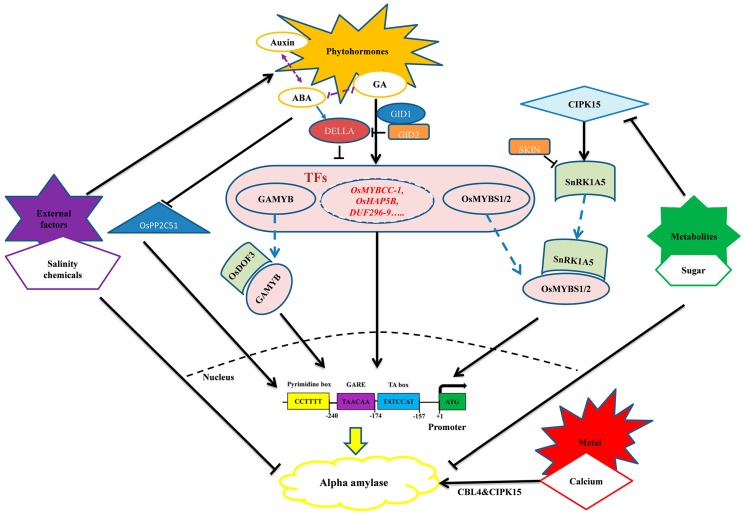
A proposed model of regulation of alpha-amylase including phytohormone, signaling, abiotic stress, and metabolites regulation. Information for developing this model was obtained from [14,51,55,83,84,85,86,87,88,89]. The transcription factors that are written in red are part of a continuing experiment and have not been verified (Unpublished data).

**Table 1 ijms-20-00450-t001:** Alpha-amylases and their corresponding group and subfamily classifications.

Gene	Subfamily	Group	Regulation	Expression Stage	Expression Site
*RAmy1A* *RAmy1B*	RAmy1	1	Phytohormones GA, ABA, sugar repression Chemical repression [50] High temperature during rice seed maturation [4]	Seed development [51] Seed germination [45]	Aleurone layer [52] Endosperm [51]
*RAMY1C*	RAmy1	1	High temperature during rice seed maturation [4]	Entire growth period *	Aleurone layer [52] All tissues *
*RAmy2A*	RAmy2	4	NF	Entire growth period *	Leaf sheath [2] All tissues *
*RAmy3A*	-	3	High temperature during rice seed maturation [4]	Seed development *	Seed maturation (endosperm)
*RAmy3B*	-	3	Chemicals repression [50]	Seed germination [45]	Aleurone layer [52]
*RAmy3C*	-	3	NF	Seed germination [45] Seed development [51]	Endosperm [51] Leaf sheath [2]
*RAmy3D*	RAmy3	2	Sugar repression [53] Calcium [54] High temperature during rice seed maturation [4]	Seed development [51] Seed germination [45]	Embryo [49] Aleurone layer [51]
*RAmy3E*	RAmy3	5	Chemical repression [50] High temperature during rice seed maturation [4]	Seed development [51] Seed germination [45]	Aleurone layer [52]
*RAmy3F*	RAmy3	6	NF	Entire growth period *	All tissues *

(-) stands for information about the gene was not found in the literature that was read. * means information from the figure in http://ricexpro.dna.affrc.go.jp/ website (Appendix A).

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
