# Peer review of "The Rice Alpha-Amylase, Conserved Regulator of Seed Maturation and Germination"

_ijms, 2019, doi:10.3390/ijms20020450_

Round 1
Reviewer 1 Report
The paper by Damaris et al. provides an extensive review of the literature concerning rice alpha-amylases. The review first provided background on rice seed and the germination process, laying out a good foundation for novice readers. This is followed by an in-depth discussion of the classification, gene structure, gene expression and regulation of rice alpha-amylases. The paper is nicely written but a number of minor suggestions can be considered before publication.
1. In figure 1, please include the meaning of all abbreviations (DAF, HAI, GA etc) for clarity. Figures and legends should stand alone from the text. In Figure 1B, remove the box with Figure 1.tif heading.
2. In figure 2B, indicate in the legends which are the barley and Arabidopsis genes for clarity. In figure 2C, there seems to be a bound molecule to the leftmost helix. Please clarify this in the legends.
3. In line 190-191, the function of the secondary carbohydrate binding sites can be expounded and these sites can be labelled in the structure shown in Figure 2C.
4. A protein sequence alignment of the 10 rice alpha-amylases can be added in the supplementary to easily compare the different sequences.
5. The information in line 227 (RAmy3A expressed at 42 DAF) does not agree with the information in table 1 (expression stage – NF). Please correct accordingly and double check data for the rest of the table.
6. Table 1 can be cited in part 5 (e.g. in line 223).
7. In figure 3, cite the major references used to develop the figure.
8. Minor grammatical and word selection issues can be found throughout the manuscript. Some examples are:
Line 12 and line 137: alpha-amylase (small first letter)
Line 42: delete “ever”
Line 83: rice seed germination, à rice seed germination. (replace comma with period)
Line 162: and RAmy3f was placed (use third person for consistency throughout the manuscript)
Line 193: cis-factor. (remove period to avoid fragmented phrases)
Line 202: response à respond
Line 217. Srytal à crystal
All greek alpha symbols can be spelled out for consistency.
Author Response
Comments and Suggestions for Authors
The paper by Damaris et al. provides an extensive review of the literature concerning rice alpha-amylases. The review first provided background on rice seed and the germination process, laying out a good foundation for novice readers. This is followed by an in-depth discussion of the classification, gene structure, gene expression and regulation of rice alpha-amylases. The paper is nicely written but a number of minor suggestions can be considered before publication.
Response: Thank you for this positive comment.
1. In figure 1, please include the meaning of all abbreviations (DAF, HAI, GA etc) for clarity. Figures and legends should stand alone from the text. In Figure 1B, remove the box with Figure 1.tif heading.
Response: The abbreviations have been spelled out in the figure legends. We have submitted the Figures separately, while the version with figures inserted in the text also was prepared according to the IJMS instructions “All Figures, Schemes and Tables should be inserted into the main text close to their first citation”.
Figure 1B have been replaced with a high-quality image and the “tif-detail box” has been removed.
2. In figure 2B, indicate in the legends which are the barley and Arabidopsis genes for clarity. In figure 2C, there seems to be a bound molecule to the leftmost helix. Please clarify this in the legends.
Response: The barley and Arabidopsis genes have been indicated in the legends as suggested for clarity.
Thank you for this reminder, this are oxygen (red) and carbon (deep blue) atoms together which constitute one of the secondary carbohydrate binding site (SBS1) and this have been indicated by an arrow in Figure 2C and described in the figure legends.
3. In line 190-191, the function of the secondary carbohydrate binding sites can be expounded and these sites can be labelled in the structure shown in Figure 2C.
Response: Thank you for this reminder, information regarding the role of these two secondary carbohydrates binding sites (line 190-198) is included in the revised manuscript.
The secondary carbohydrate binding site (SBS1) is indicated with an arrow in figure 2C and description included in the figure legends.
4. A protein sequence alignment of the 10 rice alpha-amylases can be added in the supplementary to easily compare the different sequences.
Response: Thank you for this suggestion, we have included the alignment picture as supplementary Figure 2 (line 178).
5. The information in line 227 (RAmy3A expressed at 42 DAF) does not agree with the information in table 1 (expression stage – NF). Please correct accordingly and double check data for the rest of the table.
Response: Thank you for pointing out this error. We have carefully checked the table information and included the missing information that was noted in the text but omitted in the table. (See revised table in the revised manuscript).
6. Table 1 can be cited in part 5 (e.g. in line 223).
Response: Thank you for this wonderful suggestion, table 1 have been cited in part 5.
7. In figure 3, cite the major references used to develop the figure.
Response: References used in developing the proposed model have been included in the revised manuscript.
8. Minor grammatical and word selection issues can be found throughout the manuscript. Some examples are:
Line 12 and line 137: alpha-amylase (small first letter)
Response: This has been changed.
Line 42: delete “ever”
Response: We have deleted this.
Line 83: rice seed germination, à rice seed germination. (replace comma with period)
Response: The comma has been replaced with period.
Line 162: and RAmy3f was placed (use third person for consistency throughout the manuscript)
Response: We made the change as suggested.
Line 193: cis-factor. (remove period to avoid fragmented phrases)
Response: This error was rectified.
Line 202: response à respond
Response: This was corrected in the revised manuscript.
Line 217. Srytal à crystal
Response: This was corrected in the revised manuscript.
All greek alpha symbols can be spelled out for consistency.
Response: Thank you for pointing this out, we have changed to “alpha-amylase” in the whole text
Reviewer 2 Report
Current manuscript review the role of the rice a-amylase through the seed maturation and germination process. Overall, the manuscript prepared well, and clear. Some of the previous reviews were found to deal with the related topic, however current manuscript found right frame covers the topic appropriately. There are some points require the attention of the authors as below;
- Consistent use of 'Alpha' or 'a (Greek alpha)' should be maintained through the manuscript, Which includes the title.
- Supplementary Information should be prepared in formal documentation.
Minor typos
Figure 1A; needs a detailed explanation of the color change.
Figure 1B, 3; needs a clear image. Letters are too coarse to be recognized.
line 44; D in small caps
line 65; [12]. Rice
line 94; at a time [26]
line 107; endosperm [30].
line 156; subfamilies [47]
line 174 Figure 2A
line 176; Arabidopsis in Italic
line 180 [40] conclusion
line 182, 198; no red necessary
line 193; cis in Italic
line 197; (TATCCA) [88].
line 197; cis in Italic
Figure 2; the caption is too redundant., no red necessary. D. Schemes of constructs for
line 222 and 229; alpha-amylase in Italic necessary?
line 227; DAF was abbreviated before.
line 232 [61]. The second
Author Response
Current manuscript review the role of the rice a-amylase through the seed maturation and germination process. Overall, the manuscript prepared well, and clear. Some of the previous reviews were found to deal with the related topic, however current manuscript found right frame covers the topic appropriately. There are some points require the attention of the authors as below;
Response: Thank you for the positive comments, we appreciate.
- Consistent use of 'Alpha' or 'a (Greek alpha)' should be maintained through the manuscript, Which includes the title.
Response: Thank you for pointing this out. We have changed and maintained a consistent use of “alpha-amylase” throughout the manuscript.
- Supplementary Information should be prepared in formal documentation.
Response: We prepared a formal documentation for the supplementary information.
Minor typos
Figure 1A; needs a detailed explanation of the color change.
Response: Thank you for this suggestion, we have explained the color change in the text as: Rice seeds develops from the florets of the main stem which become the immature grains (thus the initial green color of the developing seed). The mature seed is tightly packed with starch granules (hence the translucent color) while imbibition rehydrates the starch granules, making them loose and thus appearing white as a result of light refection).
Figure 1B, 3; needs a clear image. Letters are too coarse to be recognized.
Response: The figure 1B quality has been improved and the letters are visible in the new figure.
line 44; D in small caps
Response: Thank you for pointing this out, this has been changed to “d”
line 65; [12]. Rice
Response: This space error has been rectified.
line 94; at a time [26]
Response: This error is rectified.
line 107; endosperm [30].
Response: This error is rectified.
line 156; subfamilies [47]
Response: This error is rectified.
line 174 Figure 2A
Response: This error is rectified.
line 176; Arabidopsis in Italic
Response: This error is rectified.
line 180 [40] conclusion
Response: This error is rectified.
line 182, 198; no red necessary
Response: This error is rectified.
line 193; cis in Italic
Response: This error is rectified.
line 197; (TATCCA) [88].
Response: This error is rectified.
line 197; cis in Italic
Response: This error is rectified.
Figure 2; the caption is too redundant, no red necessary. D. Schemes of constructs for
Response: Figure 2 caption has been revised.
The “red” was an error and it has been rectified in the revised manuscript.
Figure 2D is a representative promoter of the alpha-amylase 1A showing the various elements involved in the expression of this gene.
line 222 and 229; alpha-amylase in Italic necessary?
Response: This was an error and it has been changed to non-Italic font
line 227; DAF was abbreviated before.
Response: Thank you for pointing this out, it was abbreviated before and we have removed “days after fertilization”.
line 232 [61]. The second
Response: This mistake has been revised.